# Effect of Heterologous Vaccination Strategy on Humoral Response against COVID-19 with CoronaVac plus BNT162b2: A Prospective Cohort Study

**DOI:** 10.3390/vaccines10050687

**Published:** 2022-04-28

**Authors:** Hakan Demirhindi, Burak Mete, Ferdi Tanir, Ertan Kara, Filiz Kibar, Salih Cetiner, Aslihan Candevir, Sukriye Ece Akti

**Affiliations:** 1Department of Public Health, Faculty of Medicine, Cukurova University, Adana 01330, Turkey; bmete@cu.edu.tr (B.M.); ftanir@cu.edu.tr (F.T.); ekara@cu.edu.tr (E.K.); seakti@cu.edu.tr (S.E.A.); 2Cukurova University Balcali Hospital Central Laboratory, Faculty of Medicine, Cukurova University, Adana 01330, Turkey; fkibar@cu.edu.tr (F.K.); scetiner@cu.edu.tr (S.C.); 3Department of Infectious Diseases and Clinical Microbiology, Faculty of Medicine, Cukurova University, Adana 01330, Turkey; acandevir@cu.edu.tr

**Keywords:** COVID-19 vaccines, seroconversion, inactivated SARS-CoV2 vaccine, BNT162 vaccine, COVID-19 vaccine booster shot, heterologous vaccination, mixed vaccination, vaccination strategy

## Abstract

This study aimed to evaluate the mixed and homogeneous application of the inactivated SARS-CoV-2 vaccine CoronaVac (CV) and the mRNA vaccine BNT162b2 (BNT). This prospective cohort study included 235 health care workers who had received two prime shots with CoronaVac. They were divided into three cohorts after the third month: Cohort-I (CV/CV); Cohort-II (CV/CV/CV); and Cohort-III (CV/CV/BNT). Anti-S-RBD-IgG and total anti-spike/anti-nucleocapsid-IgG antibody concentrations were examined in vaccinated health workers at the 1st, 3rd, and 6th months following the second dose of the vaccination. The mean age of 235 health care workers who participated in the project was 39.51 ± 10.39 (min-max: 22–64). At the end of the 6th month, no antibodies were detected in 16.7% of Cohort-I participants, and anti-S-RDB IgG levels showed a decrease of 60% compared to the levels of the 3rd month. The antibody concentrations of the 6th month were found to have increased by an average of 5.13 times compared to the 3rd-month levels in Cohort-II and 20.4 times in Cohort-III. The heterologous vaccination strategy “CoronaVac and BNT162b2 regimen” is able to induce a stronger humoral immune response and it will help remove inequalities in the developing world where CoronaVac was the initial prime.

## 1. Introduction

SARS-CoV-2 is well known as a new member of the *Betacoronavirus* (CoVs) family, which is the leading COVID-19 [1,2,3]. The vaccine research rally reached more than 50 vaccine candidates against COVID-19 at the end of 2020. There were included different platforms, such as nucleic acid, live attenuated virus, protein subunit, and viral vector vaccines [4]. In a short time, several large clinical trials have demonstrated the clinical efficacy and safety of certain COVID-19 vaccines. Each of these platforms has advantages, as well as disadvantages. As an example, nucleic acid-based vaccines such as DNA vaccines or mRNA vaccines are easy to design, but they may not be immunogenic. Moreover, evidence suggests that the mRNA vaccines are more unstable in comparison to the other types of vaccines. Viral vector-based vaccines show higher safety and are more immunogenic, while they probably have low efficiency due to the pre-existing immunity to the vector [5]. From all designed vaccines, only a few have entered the clinical trials. Although none of these vaccines has completed clinical trials, there are still many attempts in advance to develop such a vaccine [6]. Currently, the correlation between the vaccine-elicited immune response and the protection from SARS-CoV-2 infection remains unclear, and it is not known how changes in immunity could be reflected in clinical outcomes. Doria-Rose et al. showed that mRNA1273-elicited antibodies could persist for six months after the second dose in a Phase-1 trial comprising 333 healthy participants [7]. However, the durability of protection induced by COVID-19 vaccination remains to be determined. On this basis, Khoury et al. analysed seven vaccines and a convalescent cohort to design a predictive model for revealing the relationship between the observed protection and in vitro neutralization levels from COVID-19. They showed that neutralising antibody levels are highly predictive of immune protection. For achieving a 50% protection, the neutralisation level was estimated to be approximately 20.2% of the average convalescent level (95% CI, 14–28) and 3% (95% CI, 0.7–13) for 50% protection from severe infection, and that the decay of neutralizing titres in vaccinated subjects over the first 3–4 months after vaccination was, at least, as rapid as the decay observed in the convalescent subjects. Furthermore, modelling the neutralization titres decay over the first 250 days after immunisation showed a significant loss in protection from SARS-CoV-2 infection while protection from severe disease would be largely preserved [8].

Heterologous prime-boost vaccination, which defines mixing different vaccine types, is a strategy that originated in the 1990s to be used against HIV infection by aiming to induce both T-cell and B-cell immunity. The 30-year cumulative result of these vaccine platforms, which included DNA, mRNA, and viral vectors, is what rendered the development of COVID-19 vaccines so quickly possible (reminding that this does not include DNA vaccines). Mixing vaccines is based on the idea of presenting the antigen to the immune system in a different way. This approach is particularly important for viral vector vaccines against COVID-19, such as Sputnik V, Johnson & Johnson, CanSino Biologics, and AstraZeneca vaccines that use a replication-deficient adenovirus as a vector. However, the immune system may attack the vector virus itself, rendering the vaccine ineffective. To avoid this, it is accepted that third shots would be shifted to mRNA- or protein-based vaccines following the prime series with vector vaccines. The main aim is effective and long-lasting protection [9]. There are several studies like Com-COV (1-2-3) study or COV-Boost study in the United Kingdom, that compareimmune responses and any side effects of the combination of the prime shots with one type and a booster shot with another type. They comprise AstraZeneca, Pfizer & BioNTech, Moderna, and Novavax vaccines [10,11]. The next generation of vaccines are expected to be effective against several coronavirus variants with a resulting broader immunity, longevity of response, and a good safety profile. [9].

We aimed in this study to evaluate whether the mixed application of the inactivated vaccine and the mRNA vaccine BNT162b2 had a superiority in antibody response and safety compared to homologous vaccine application.

## 2. Materials and Methods

### 2.1. Study Design and Participants

This was a prospective cohort study and constituted the third step of a vaccine efficacy and safety project. It was carried out at Cukurova University (Adana, Turkey) in September 2021 and included health care workers who had been vaccinated with two doses of the inactivated SARS-CoV-2 vaccine in the context of a public vaccination program by the Turkish Ministry of Health (TR-MoH, Ankara, Turkey). The health care workers constituted the priority group in the vaccination target population. The incidence of adverse events in the vaccine cohort was determined to be 19% after the administration of 3 μg (corresponding to 600 SU) of the inactivated SARS-CoV-2 vaccine during Phase-2 clinical trials [12]. The minimum sample size was calculated as 220 participants by assuming type-1 error as 0.05 and type-2 error as 0.20 considering reported adverse events rates. The participants were randomly selected from a list of 3000 health care workers. Two substitution lists were also prepared by randomisation. A total number of 282 health care workers participated in the first step of the study performed at the end of the first month after vaccination. This number decreased to 272 in the second step of the study performed at the end of the third month after the second dose of the vaccine when 10 participants resigned from the study of their own free will. The third step of the study included 235 participants and the calculations were based on the results of these participants.

Blood samples were taken after the participants had been informed about the study and had signed an informed consent form. They answered a questionnaire form about sociodemographic characteristics, natural COVID-19 infection history, and vaccination and adverse events.

### 2.2. Vaccine Information

#### 2.2.1. Inactivated SARS-CoV-2 Vaccine by Sinovac (CoronaVac^TM^)

The vaccine administered to health care workers by the TR-MoH is “inactivated SARS-CoV-2 vaccine (CoronaVac^TM^)”, with aluminium hydroxide, developed by Sinovac Biotech Ltd., Life Sciences Lab., Beijing, China. The vaccine (that will be named shortly as CV) was administered intramuscularly in the deltoid region of the upper arm with a dosage of 3 μg/0.5 mL. The two doses were administered 28 days apart. The vaccines were transferred, stored, and administered following cold-chain principles already in use by, and therefore familiar to, health institutions performing vaccinations.

#### 2.2.2. BNT162b2 mRNA Vaccine by Pfizer & BioNTech (Comirnaty^®^)

The vaccine BNT162b2 (Comirnaty^®^) produced by BioNTech Manufacturing GmbH, Germany is a nucleoside-modified messenger-RNA (mRNA) encapsulated in lipid nanoparticles (LNP), which enables the delivery of the RNA into host cells to allow expression of the SARS-CoV-2 spike (S) antigen. The vaccine (that will be named shortly as BNT) elicits both neutralising antibody and cellular immune responses to the S antigen, which may contribute to protection against COVID-19 disease. The mRNA of the vaccine is a highly purified single-stranded, 5′-capped mRNA produced using a cell-free in vitro transcription from the corresponding DNA templates, encoding the S protein of SARS-CoV-2. This vaccine is a white-to-off-white frozen suspension provided as a multiple-dose vial and must be diluted before use. One vial (0.45 mL) contains 6 doses of 0.3 mL after dilution. One dose (0.3 mL) contains 30 micrograms of COVID-19 mRNA vaccine (embedded in lipid nanoparticles) [13].

#### 2.2.3. Mixed Vaccine Administration

In June 2021, the Turkish Republic MoH declared the introduction of an additional shot of a third dose to health care workers and elderly who had previously received two doses of CoronaVac and to the individuals who wished to receive it due to some international travel requirements. All individuals were given the right to choose between CV and BNT vaccines of their free will.

### 2.3. Immune Response Assessments

The project aimed to measure anti-SARS-CoV-2 S-RBD (anti-S-RBD) immunoglobulin G (IgG), total anti-spike and anti-nucleocapsid immunoglobulin IgG antibody concentrations in the same individuals repetitively, i.e., at the end of the first month (first step of the study), in the third month (second step), and in the sixth month (third step) after the administration of the second dose of CV. As the TR-MoH introduced the third dose (booster) in the vaccination schedule in June 2021, the evaluation of the same antibodies following this booster dose was also included in this presented third step of the study, performed at the end of the sixth month. As the individuals were given the right to choose, the third (booster) dose consisted of either CV or BNT. Therefore, all participants were divided into three cohorts: -Cohort-I (CV/CV) included those who received only two doses of CoronaVac (*n* = 50);-Cohort-II (CV/CV/CV) included those who received three doses of CoronaVac (*n* = 17);-Cohort-III (CV/CV/BNT) included those who received two doses (prime) of CoronaVac followed by the third dose (booster) with BNT162b2 (*n* = 168).

About 5 mL of blood samples were collected into biochemistry tubes with vacuum gel. The sera were extracted by centrifugation at 3000 *g* for 10 min and kept at 2–8 °C for 1–3 days. Test calibrators and controls were performed first. After the control results were observed to be within the expected ranges, the samples were tested by trained experts in the accredited (by the Joint Commission International (JCI) since 2006) Central Laboratory of Cukurova University Balcali Hospital, Adana, Turkey with the MAGLUMI 2000 series fully automated chemiluminescence immunoassay analyser (CLIA) (Snibe Diagnostics, Shenzen New Industries Biomedical Engineering Co. Ltd., Shenzhen, China) [14].

The test kit for the determination of antibodies was MAGLUMI^®^ SARS-CoV-2 S-RBD IgG (CLIA) (Cat. #130219017M) (Snibe Diagnostics, Shenzen New Industries Biomedical Engineering Co. Ltd., Shenzhen, China). The blood samples stored at 2–8 °C were brought to room temperature before the analysis on the working day and examined collectively after all participants were sampled. The SARS-CoV-2 S-RBD IgG (CLIA) assay is an indirect chemiluminescence immunoassay. The sample, buffer, and magnetic microbeads coated with S-RBD recombinant antigen are mixed thoroughly and incubated, forming immune complexes. After precipitation in a magnetic field, the supernatant is decanted, and a wash cycle is performed. Then, ABEI labelled with anti-human IgG antibody is added and incubated to form complexes. After precipitation in a magnetic field, the supernatant is decanted, and a wash cycle is performed. Subsequently, the Starter 1 + 2 are added to initiate a chemiluminescent reaction. The light signal is measured by a photomultiplier as relative light units (RLUs), which is proportional to the concentration of SARS-CoV-2 S-RBD IgG presented in the sample. 

The analyser automatically calculates the numerical output in each sample by means of a calibration curve, which is generated by a two-point calibration master-curve procedure. The results are expressed in absorbance units (AU/mL). The results are reported to the end-user as “Reactive” and “Non-Reactive”. A result less than 1.00 AU/mL (<1.00 AU/mL) is considered to be non-reactive. A result greater than or equal to 1.00 AU/mL (≥1.00 AU/mL) is considered to be reactive [14]. The SARS-CoV-2 S-RBD IgG test is an indirect CLIA and has a high correlation with VNT50 titres (R = 0.712), where VNT stands for “Virus Neutralization Test”, which is a gold standard for quantifying the titre of neutralising antibodies (nAbs) for a virus [15]. The test is only for use according to the Food and Drug Administration’s Emergency Use Authorization [16].

### 2.4. Statistical Analyses

Data were examined using the SPSS 22 statistical analyses package (2013, IBM, New York, NY, USA). Following normality testing (Kolmogorov–Smirnov), data were analysed by Freidman and Kruskal Wallis tests. A value of *p* < 0.05 was considered significant.

## 3. Results

The mean age of 235 health care workers who participated in the third step of our study was 39.51 ± 10.39 (between 22 and 64). All participants received two doses of CoronaVac (CV) one month apart at the start of the study. At the end of the sixth month following the second dose, 21.3% of the participants reported to have received only two doses of CoronaVac in Cohort-I (CV/CV), 7.2% reported to have received three doses of CoronaVac in Cohort-II (CV/CV/CV), and 71.5% reported to have received two doses of CoronaVac followed by the third dose with BNT162b2 (BNT) vaccine in Cohort-III (CV/CV/BNT).

Some participants reported a COVID-19 infection history before the first dose in all three cohorts (6, 5, and 26, respectively). However, those who reported the infection after the second dose were observed in the CV/CV cohort (two people) and the CV/CV/BNT cohort (seven people), the latter being observed between the second and third doses. There was not any infection history between the second and third doses in the CV/CV/CV cohort.

When the participants who had received the third dose were considered, the mean of days passed between the second and third dose administrations was 148.3 days for the CV/CV/CV cohort and 142.8 days for the CV/CV/BNT cohort. Eight participants were not included in the analyses, because the interval between the third shot and the blood sampling for our study was shorter than 14 days. This interval was found to range between 18 and 71 days (Table 1).

At the end of the sixth month, no antibodies could be detected in 16.7% of the individuals in the CV/CV cohort and the level of anti-S-RDB IgG was found to be significantly decreased by a mean of 60% compared to the third month. The anti-S-RDB IgG was found detectable in all individuals in the CV/CV/BNT cohort. The level of antibodies in the sixth month was found to have increased by an average of 5.13 times in the CV/CV/CV cohort and 20.4 times in the CV/CV/BNT cohort. Anti-S-RBD IgG levels were found to be significantly higher in the CV/CV/BNT group than in the CV/CV/CV group or CV/CV group. In other words, the participants who received the BNT vaccine as the third dose presented higher IgG levels than all other cohorts (Table 2).

When the total anti-spike/anti-nucleocapsid IgG antibody level in the sixth month was compared to that of the third month, it was found to be undetectable in 71.4% in the CV/CV cohort with a decrease of 91%, to have increased by an average of 3.5 times in the CV/CV/CV cohort, and to have decreased by 87% in the CV/CV/BNT cohort (Table 3).

The distribution of anti-S-RBD IgG levels according to months and vaccine cohorts are presented in Figure 1.

When the anti-S-RBD IgG levels were compared between the vaccine cohorts, it was found that they had a statistically significant decrease of 51% among participants without any history of COVID-19 infection in the CV/CV cohort. However, among the participants who had the infection before the first dose or after the second dose in the same CV/CV cohort, the reduction of antibody levels in the sixth month was not statistically significant.

In the CV/CV/CV cohort, the antibody levels were found to have increased 6.4 times in those who had COVID-19 history. However, the increase was 1.5 times among those who had the infection before the start of the vaccinations. 

In the CV/CV/BNT cohort, the level of antibodies increased 23 times among participants without COVID-19 history, but 2.3 times who got infected before the vaccinations, and 8.8 times in those who got infected after the second dose, but before the third dose of the vaccine. The decrease was observed to be at a slower rate in individuals with a positive history of natural COVID-19 infection (Table 4).

The anti-S-RBD IgG antibody levels were found to have decreased in all age groups of the CV/CV cohort, while, in contrast, to have increased in all age groups of the CV/CV/CV cohort and the CV/CV/BNT cohort. The increase was higher in the CV/CV/BNT cohort than the CV/CV/CV cohort (Table 5).

The adverse events were found to be more frequent in participants who received the mRNA vaccine as the third dose of the vaccination (Table 6).

## 4. Discussion

Ensuring an effective and durable immune response is very important in reaching the herd immunity threshold essential for ending the pandemic. In the face of limited vaccine supply, a mixed-vaccine strategy was proposed to protect as many people as possible. On this basis, a 13-month study was launched by Shaw et al. on 4 February 2021, to determine the effects of using different vaccines for each dose in different testing arms: (1) the ChAdOx1 nCov-19 vaccine (AZD1222) by AstraZeneca for the first dose followed by the BNT162b2 (BNT) for the second dose 28 days later; (2) the AZD1222 for the first dose, followed by the BNT for the second dose 12 weeks later; (3) the BNT vaccine for the first dose, followed by the AZD1222 vaccine for the second 28 days later; and (4) the BNT vaccine for the first dose, followed by the AZD1222 for the second dose 12 weeks later. The initial report showed that there was an increase in systemic reactogenicity after the boost dose, as reported by participants in heterologous vaccine schedules in comparison to homologous vaccine schedules [17]. Another trial is being performed to examine the safety and immunogenicity of combining the AZD1222 and rAd26-S (Gam-COVID-Vac; Sputnik V) vaccines, which are both adenovirus vector vaccines. One group received AZD1222 on day one followed by rAd26-S on day 29 and the second group received the same vaccines in the reverse order. The preliminary Phase-2 results of the study revealed a four-fold or greater increase in neutralising antibody levels compared to the baseline [18]. In a study conducted in Spain by Borobia et al. between 24 April and 30 April 2021, 676 individuals were randomized (450 in the intervention and 226 in the control group) at five sites in Spain; among them, 663 (441 and 222, respectively) completed the study up to day 14 (mean age 44; 56.5% female). In the intervention group, geometric mean titres (GMT) of IgG-RBD were found to increase from 71.46 BAU/mL (95% CI, 59.84–85.33) at baseline to 7756.68 (95% CI, 7371.53–8161.96) at day 14 (*p* < 0.0001). IgG against trimeric spike protein increased from 98.4 (95% CI, 85.69–112.99) to 3684.87 (95% CI, 3429.87–3958.83). All of the participants exhibited neutralising antibodies 14 days after BNT administration, in comparison to 34.1% at enrolment. A four-fold increase in cellular immune response was also observed. Reactions were predominantly mild (68.3%) or moderate (29.9%), and consisted of injection site pain (88.2%), induration (35.5%), headache (44.4%), and myalgia (43.3%). No serious adverse events were reported. BNT given as a second dose in individuals primely vaccinated with AZD1222 induced a robust immune response with an acceptable and manageable reactogenicity profile [19]. In a study at Charité–Universitätsmedizin Berlin in Germany including 340 health care workers immunised between 27 December 2020 and 21 May 2021, it was aimed to assess the reactogenicity and immunogenicity of heterologous prime-boost immunisations of AZD1222 followed by BNT compared to homologous BNT/BNT immunisation. The heterologous AZD1222/BNT booster vaccination was overall well-tolerated and reactogenicity was largely comparable to the homologous BNT/BNT vaccination. Systemic reactions were most frequent after prime immunisation with AZD1222 (86%, 95%CI, 79–91), and less frequent after the homologous BNT/BNT (65%, 95%CI, 56–72) or heterologous AZD1222/BNT booster vaccination (48%, 95%CI, 36–59). Serum antibody responses and T-cell reactivity were found to strongly increase after both homologous and heterologous boosts, and immunogenicity was overall robust and comparable between both regimens in this cohort, with a slightly increased S1-IgG avidity and T-cell response following the heterologous booster immunisation. Evidence of rare thrombotic events associated with AZD1222 has led to the recommendation of a heterologous booster with mRNA vaccines for certain age groups in several European countries, despite a lack of robust safety and immunogenicity data for this vaccine regimen. This interim analysis provided evidence that the currently recommended heterologous AZD1222/BNT immunisation regimen with 10–12-week vaccine intervals was well tolerated and slightly more immunogenic compared to the homologous BNT/BNT vaccination with three-week vaccine intervals. The heterologous prime-boost immunisation for COVID-19 was concluded to be generally applicable to optimise logistics, improve immunogenicity, and mitigate potential intermittent supply shortages for individual vaccines [20]. In three recent studies, researchers reported that following one dose of AZD1222 with a dose of the BNT had produced strong immune responses, as measured by blood tests. Two of the studies even suggested that the mixed-vaccine response would be at least as protective as two doses of the BNT vaccine, one of the most effective COVID-19 vaccines. Only a few of the potential vaccine combinations were tested. However, if mixing vaccines proves safe and effective, it could speed up the effort to protect billions of people [21].

In our study, the heterologous use of the inactivated vaccine and mRNA vaccine was compared with homologous vaccination in terms of humoral immune response and adverse reactions. We found that the heterologous administration of CV and BNT vaccines produced a stronger immune response than the homologous regimen. At the end of the sixth month, antibodies could not be detected in 16.7% of the individuals who received two doses of the inactivated vaccine (CV/CV), and the level of anti-S-RDB IgG decreased by 60% on average compared to the third-month levels. After the third dose of CV the antibody level in the sixth month increased by an average of 5.13 times compared to the third month, and 20.4 times in those who received the BNT vaccine. It should be noted that the fact that there is a higher proportion of anti-S-RBD response in those re-immunised with BNT does not imply that these individuals are more protected, since this study does not assess protection, but levels of antibodies against the antigen. In terms of adverse events, it was observed that 97.5% of the side effects were in the heterologous vaccine group, while 2.5% of them were in the homologous vaccine group. Similarly, Pérez-Then et al. reported that heterologous CoronaVac prime followed by a BNT booster regimen induced elevated virus-specific antibody levels and potent neutralization activity against the ancestral virus and Delta variant, resembling the titres obtained after two doses of mRNA vaccines. They also pointed out that CV/CV/BNT resulted in a 1.4-fold increase in neutralization activity against Omicron, compared to BNT/BNT regimen [22].

The limitations of the study can be noted as cellular immunity response was not assessed and the sampling size was not large. As the study was observational and the selection of the third dose was left to the decision of the individuals (without any intervention by researchers), the proportional parity in the number of participants in three cohorts could not be provided.

## 5. Conclusions

In conclusion, our study revealed that the heterologous administration of inactivated SARS-CoV-2 (CoronaVac^TM^) prime followed by BNT162b2 (Comirnaty^®^) booster regimen produced a stronger immune response than the homologous inactivated vaccine regimen, but more adverse events were observed. About 17% of people who received only two doses of the CoronaVac vaccine lacked antibodies. One of the important results of the study is that a booster dose is needed at the end of the sixth month. Mixed (heterologous) vaccination with the inactivated SARS-CoV-2 vaccine and mRNA vaccine seems to be more effective in providing a stronger antibody response. Mixed vaccine strategies need more evidence. If mixed vaccine strategies prove to be effective, vaccination programs will be more flexible, thereby reducing the negative impact of supply chain disruptions. Our findings also have immediate implications for multiple countries that previously used the CoronaVac regimen.

## Figures and Tables

**Figure 1 vaccines-10-00687-f001:**
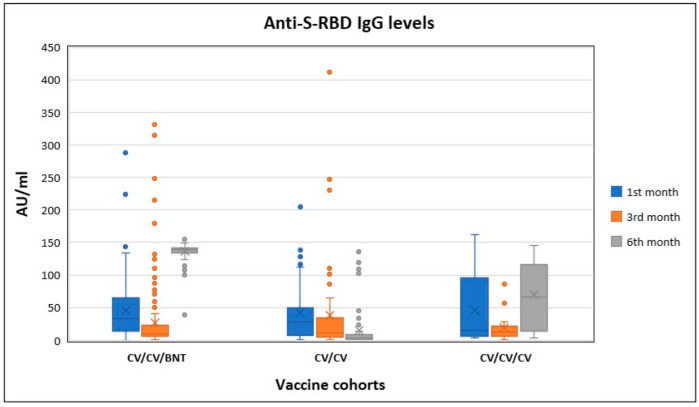
The distribution of anti-S-RBD IgG levels according to months and vaccine cohorts.

**Table 1 vaccines-10-00687-t001:** Sociodemographic characteristics of the participants according to vaccine cohorts.

	Vaccine Cohorts		
	*n* (%) or Mean ± Standard Deviation (Min-Max)		
Characteristics	Cohort-I	Cohort-II	Cohort-III
Sex			
Male	24 (24.5)	7 (7.1)	67 (68.4)
Female	26 (19.0)	10 (7.3)	101 (73.7)
Chronic disease present			
Yes	11 (18.0)	4 (6.6)	46 (75.4)
No	38 (22.0)	13 (7.5)	122 (70.5)
History of COVID-19 infection			
Never	42 (22.0)	12 (6.3)	137 (71.7)
Before the 1st dose	6 (16.2)	5 (13.5)	26 (70.3)
After the 2nd dose	2 (28.6)	0 (0)	7 (71.4)
Age	35.66 ± 8.13	39.29 ± 8.18	40.67 ± 10.94
Interval between		148.35 ± 15.32	142.85 ± 10.48
2nd and 3rd doses (days)		(136–180)	(136–186)
Interval between		39.52 ± 10.50	45.92 ± 12.47
2nd dose and blood sampling (days)		(18–50)	(18–71)

**Table 2 vaccines-10-00687-t002:** Anti-S-RBD IgG levels according to months and vaccine cohorts.

		Anti-S-RBD IgG (AU/mL)			Antibody Proportions			Humoral Immune Response
Vaccine Cohorts		1st Month	3rd Month	6th Month	*p*	6th/1st ^a^	6th/3rd ^b^	N/R (%) ^c^
Cohort-I *n* = 42	Mean	44.38	23.47	11.14				
	S.D.	49.84	40.51	24.97	**<0.001**	0.64	0.40	16.7/83.3
	Median	27.57	7.53	2.66				
Range		203.73	229.62	119.14				
Cohort-II *n* = 12	Mean	46.61	15.02	76.18				
	S.D.	55.53	15.65	53.28	**0.003**	3.53	5.13	0/100.0
	Median	13.99	8.61	85.99				
	Range	158.16	55.46	139.06				
Cohort-III *n* = 137	Mean	42.34	17.59	136.82				
	S.D.	37.73	27.17	11.00	**<0.001**	17.00	20.44	0/100.0
	Median	29.63	9.21	139.20				
	Range	146.10	214.71	116.40				

The individuals with a history of COVID-19 before or after the vaccinations were excluded from the analyses. ^a^ IgG level changes as a proportion of 6th-month to 1st-month levels. ^b^ IgG level changes as a proportion of 6th-month to 3rd-month levels. ^c^ N: non-Reactive; R: reactive. Significant *p* values are highlighted in bold.

**Table 3 vaccines-10-00687-t003:** Total anti-spike/anti-nucleocapsid IgG levels according to months and vaccine cohorts.

Total Anti-Spike/Anti-Nucleocapsid IgG						Antibody Proportions		Humoral Immune Response
Vaccine Cohort		1st Month	3rd Month	6th Month	*p*	6th/1st ^a^	6th/3rd ^b^	N/R (%) ^c^
Cohort-I *n* = 42	Mean	30.26	22.72	1.91				
	S.D.	38.68	45.32	3.77	**<0.001**	0.30	0.09	71.4/28.6
	Median	11.98	3.39	0.054				
	Range	143.02	251.08	15.80				
Cohort-II *n* = 12	Mean	12.81	9.92	22.01				
	S.D.	11.36	12.82	28.64	**0.035**	4.15	3.58	16.7/83.3
	Median	10.60	2.85	9.42				
	Range	35.05	36.69	89.43				
Cohort-III *n* = 137	Mean	27.74	19.01	1.82				
	S.D.	44.08	49.10	5.64	**<0.001**	0.10	0.13	75.2/24.8
	Median	10.88	3.82	0.062				
	Range	367.68	407.32	40.72				

The individuals with a history of COVID-19 before or after the vaccinations were excluded from the analyses. ^a^ IgG level changes as a proportion of 6th-month to 1st-month levels. ^b^ IgG level changes as a proportion of 6th-month to 3rd-month levels. ^c^ N: Non-Reactive, R: Reactive. Significant *p* values are highlighted in bold.

**Table 4 vaccines-10-00687-t004:** Anti-S-RBD IgG levels according to months, vaccine cohorts, and COVID-19 history.

			Anti-S-RBD IgG Levels				Antibody Proportions	
Vaccine Cohorts	COVID-19 History		1st Month	3rd Month	6th Month	*p*	6th/1st ^a^	6th/3rd ^b^
Cohort-I	No (*n* = 42)	Mean	44.38	23.47	11.14			
		S.D.	49.84	40.51	24.97	**<0.001**	0.47 ^e^	0.39 ^f^
		Median	27.57	7.53	2.66			
		Range	203.73	229.62	119.14			
	Yes, before the 1st dose	Mean	31.71	42.23	13			
	(*n* = 6)	S.D.	14.44	34.36	10.56	0.135	0.49	0.47
	90.0 ± 46 (30–150) ^c^	Median	33.15	26.45	8.51			
	days before the 1st dose	Range	30.51	85.11	28.17			
	Yes, after the 2nd dose	Mean	33.03	329.7	118.9			
	(*n* = 2) ^d^	S.D.	24.52	116.81	23.75	0.135	4.6	0.37
	55.0 ± 21.21 (40–70) ^d^	Median	33.03	329.7	118.9			
	days after the 2nd dose	Range	34.68	165.2	33.6			
Cohort-II	No	Mean	46.61	15.02	76.18			
	(*n* = 12)	S.D.	55.53	15.65	53.28	**0.003**	4.02 ^g^	6.44 ^h^
		Median	13.99	8.61	85.99			
		Range	158.16	55.46	139.06			
	Yes, before the 1st dose	Mean	42.03	28.95	49.83			
	(*n* = 5)	S.D.	44.03	32.22	44.84	0.779	2.07	1.51
	98.60 ± 54.17 (43–180) ^c^	Median	14.68	19.4	45.12			
	days before the 1st dose	Range	95.35	81.26	101.51			
Cohort-III	No	Mean	42.34	17.59	136.82			
	(*n* = 137)	S.D.	37.73	27.17	11	**<0.001**	19.68 ^j^	23.19 ^k^
		Median	29.63	9.21	139.2			
		Range	146.1	214.71	116.4			
	Yes, before the vaccinations	Mean	66.28	42.41	137.7			
	(*n* = 26)	S.D.	69.09	45.71	4.74	**<0.001**	5.33	8.89
	130.2 ± 91.31 (30–330) ^c^	Median	38.48	22.02	137.75			
	days before the 1st dose	Range	280.32	175.18	18.3			
	Yes, between the 2nd and 3rd dose	Mean	39.91	188	134.22			
	(*n* = 5)	S.D.	16.4	153.94	19.28	0.247	4.02	2.94
	88.0 ± 51.67 (60–180) ^d^	Median	33.41	248	142.7			
	days after the 2nd dose	Range	38.95	319.73	45.7			

For all of the participants, the history of natural COVID-19 infection dates either before the first dose or between the second and third dose of vaccinations ^c,d^. ^a^ IgG level changes as a proportion of 6th-month to 1st-month levels. ^b^ IgG level changes as a proportion of 6th-month to 3rd-month levels. ^c^ Time interval of COVID-19 infection before the vaccinations in days (mean ± standard deviation, min–max). ^d^ Time interval of COVID-19 infection between the second and third dose of the vaccinations in days (mean ± standard deviation, min–max). *p* values in post-hoc analyses: ^e^
*p* < 0.001, ^f^ *p* < 0.001, ^g^ *p* = 0.264, ^h^ *p* = 0.002, ^j^ *p* < 0.001, and ^k^ *p* < 0.001. Significant *p* values are highlighted in bold.

**Table 5 vaccines-10-00687-t005:** Anti-S-RBD IgG levels in vaccine cohorts according to the age groups.

Vaccine Cohorts	Age (Years)		Anti-S-RBD IgG Levels				Antibody Proportions	
			1st Month	3rd Month	6th Month	*p*	6th/1st ^a^	6th/3rd ^b^
Cohort-I	20–29 (*n* = 12)	Mean	42.91	16.11	4.82			
		S.D.	42.30	22.92	6.59	**<0.001**	0.31	0.34
		Median	35.61	8.08	2.15			
		Range	129.93	84.31	24.01			
	30–39 (*n* = 17)	Mean	67.04	39.44	22.10			
		S.D.	61.91	57.35	36.72	**<0.001**	0.27	0.47
		Median	48.64	25.63	6.26			
		Range	202.24	229.62	119.14			
	40–49 (*n* = 11)	Mean	15.17	9.77	2.74			
		S.D.	11.96	11.37	2.47	**<0.001**	1.04	0.35
		Median	9.97	4.67	1.91			
		Range	34.04	38.40	8.21			
	50 and older (*n* = 2)	Mean	21.36	7.21	2.19			
		S.D.	14.70	3.84	1.07	0.135	0.11	0.30
		Median	21.36	7.21	2.19			
		Range	20.80	5.44	1.52			
Cohort-II	20–29 (*n* = 2)	Mean	98.43	32.31	94.41			
		S.D.	89.32	33.96	53.71	0.223	1.20	4.57
		Median	98.43	32.31	94.41			
		Range	126.33	48.04	75.97			
	30–39 (*n* = 5)	Mean	28.70	10.52	99.12			
		S.D.	42.42	8.422	52.96	**0.018**	7.02	10.65
		Median	12.89	9.99	105.30			
		Range	99.98	20.39	135.69			
	40–49 (*n* = 2)	Mean	102.25	20.42	94.14			
		S.D.	18.30	9.71	36.56	0.223	0.96	5.67
		Median	102.25	20.42	94.14			
		Range	25.89	13.74	51.71			
	50 and older (*n* = 3)	Mean	4.80	5.90	13.81			
		S.D.	1.46	1.09	9.43	0.264	2.94	2.59
		Median	4.62	5.86	10.56			
		Range	2.92	2.18	18.00			
Cohort-III	20–29 (*n* = 26)	Mean	49.34	22.95	138.94			
		S.D.	40.50	29.62	4.81	**<0.001**	6.19	15.47
		Median	34.14	10.39	139.60			
		Range	140.81	121.32	17.00			
	30–39 (*n* = 30)	Mean	44.10	14.74	137.29			
		S.D.	36.72	17.40	7.14	**<0.001**	7.64	18.98
		Median	29.95	9.08	138.10			
		Range	130.97	85.35	34.90			
	40–49 (*n* = 43)	Mean	45.97	13.72	137.92			
		S.D.	40.64	16.13	7.09	**<0.001**	8.92	23.34
		Median	25.56	9.07	139.90			
		Range	127.45	96.56	40.90			
	50 and older (*n* = 38)	Mean	32.05	20.56	133.76			
		S.D.	32.12	39.18	17.82	**<0.001**	50.61	30.49
		Median	17.88	7.93	138.15			
		Range	143.20	214.71	109.80			

^a^ IgG level changes as a proportion of 6th-month to 1st-month levels. ^b^ IgG level changes as a proportion of 6th-month to 3rd-month levels. Significant *p* values are highlighted in bold.

**Table 6 vaccines-10-00687-t006:** The adverse events after the third dose of the vaccination.

Adverse Events	Vaccine Administered as the Third Dose [*n*(line %)]	
	Inactivated Vaccine (*n*:17)	mRNA Vaccine (*n*:168)
Total	3 (2.5)	116 (97.5)
Rash at the injection site	0 (0.0)	13 (100.0)
Pain at the injection site	1 (0.9)	113 (99.1)
Swelling at the injection site	1 (5.0)	19 (95.0)
Itching at the injection site	-	9 (100.0)
Hypoesthesia at the injection site	-	2 (100.0)
Induration at the injection site	-	15 (100.0)
Numbness in the vaccinated arm	-	18 (100.0)
Weakness	2 (3.5)	55 (96.5)
Fatigue	-	48 (100.0)
Fever	-	19 (100.0)
Shake	-	8 (100.0)
Chest pain	-	5 (100.0)
Diarrhoea	-	2 (100.0)
Nausea	-	7 (100.0)
Vomiting	-	2 (100.0)
Headache	1 (3.4)	28 (96.6)
Dizziness	1 (3.7)	26 (96.3)
Vertigo	-	10 (100.0)
Muscle pain	-	30 (100.0)
Back pain	-	21 (100.0)
Joint pain	-	23 (100.0)
Cough	-	2 (100.0)
Sore throat	-	2 (100.0)
Dyspnea	-	3 (100.0)
Papule	-	-
Abdominal pain	-	1 (100.0)
Loss of appetite	-	6 (100.0)
Palpitation	-	6 (100.0)
Anosmia	-	1 (100.0)
Loss of taste	-	-
Rash in the skin/mucosa	-	-
Numbness in the tongue	-	-
Syncope	-	-
Increased blood pressure	-	1 (100.0)
Decreased blood pressure	-	-
Allergic reaction/urticaria	-	-
Anaphylaxis/anaphylactoid reaction	-	-
Neurological complications	-	-
Lymphadenopathy	-	3 (100.0)

## Data Availability

The data presented in this study are available on request from the corresponding author. The data are not publicly available due to personal data protection regulations.

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
