# Peer review of "Effect of Heterologous Vaccination Strategy on Humoral Response against COVID-19 with CoronaVac plus BNT162b2: A Prospective Cohort Study"

_vaccines, 2022, doi:10.3390/vaccines10050687_

Round 1
Reviewer 1 Report
The authors examine antibody responses following priming with the inactivated CoronaVac anti-SARS-CoV-2 vaccine from China and cross boosting with an mRNA vaccine. Little has been reported on this vaccine.
The data are clearly presented. If authors want to add more insights; they can evaluate the levels of neutralizing antibody and/or the levels of protection from symptomatic disease in a subsequent report.
Relatively little is known about the CoronaVac vaccine, so this bit of information is helpful and merits publication.
Results do not comment on protection from infection or hospitalization.
Author Response
A point-by-point response to the reviewer’s comments is provided as an attachment. The added/modified parts have been highlighted in yellow in the revised mansucript.
The added/modified parts have been highlighted in yellow in the revised mansucript. Question 1 The authors examine antibody responses following priming with the inactivated CoronaVac anti-SARS-CoV-2 vaccine from China and cross boosting with an mRNA vaccine. Little has been reported on this vaccine. The data are clearly presented. If authors want to add more insights; they can evaluate the levels of neutralizing antibody and/or the levels of protection from symptomatic disease in a subsequent report. Relatively little is known about the CoronaVac vaccine, so this bit of information is helpful and merits publication.
Reply 1 We would first like to thank reviewer 1 for the time and effort.
As the reviewer suggested, the protective effect of the CoronaVac will be evaluated at the end of the first year. The data relating to the infection control in both the prior to the vaccination and after or inter-vaccination periods have been presented in the manuscript. To summarise all participants who were infected have been recovered without any need for hospitalisation.
Reviewer 2 Report
Tamarindi et al. present the results of a study with a group of 235 health workers that sought to evaluate the humoral immune response against SARS-CoV2 after a first immunization with an inactivated virus vaccine followed by three booster-immunization conditions: 1, control without booster dose (cohort 1), 2. booster with inactivated virus vaccine (cohort 2), 3. booster with mRNA vaccine (cohort 3).
The main problem with the study derives from the fact that the selection of the type of vaccine used in the booster dose was left to the choice of the study participants themselves. As it was not a random selection, a great disproportion was generated in the number of components of each of the cohorts. Considering only the patients who did not suffer from the infection, cohort 2 is made up of only 12 participants compared to 137 in cohort 3 (11 times more!). This disproportion of participants must be considered as a strong limitation and statistical methods must be applied to guarantee that the conclusions of the study are adequate.
Precisely because of such a limited number, the differences described in the increase in antibody titers between groups 2 and 3 in lines 245-252 of the manuscript (6.4 times in group 0 2 vs 23 times in group 3) perhaps may not be significant. In the manuscript only the facts are described but it is not explained whether these differences observed are significant or not, please compare both groups and provide the value of p. It is important because if they are not significant, it could be deduced that there are no differences between the vaccination strategies.
Discussion: it should be explained as a limitation that cellular immunity (TH1-mediated) was not assessed in the study, which is the most effective type of immune response against SARS CoV2. Other similar studies, such as the one cited in the reference, assess* both types of response.
References to vaccines must be unified and always named in the same way. In line 284 it is defined that the AstraZeneca vaccine will be named AZD1222 but later in lines 314, 315, 317, 319 and 333 it is named ChAdOx1. Pag 364 the concept of Immunogenicity.
Misuse of the word “immunogenicity”. The authors try to explain that the antibody response is greater after boosting with the BNT vaccine and to this end they say that providing “stronger immunogenicity”. The term immunogenicity is inappropriate for this situation. Immunogenicity is the ability of an antigen to provoke an immune response. Immunogenicity is a binary term (like gestation) a molecule is immunogenic or it is not. What varies, depending on the people and the immunization strategies, is the intensity of the immune response against the antigen (immunogen and antigen are two different concepts). Antigen is the target of the immune attack. Please replace the corresponding terms with "...in providing stronger antibodies response."
Methods. In the description of the cohorts (lines 156-159), the number of people that made up that cohort must be indicated. Although that number is discussed later, the reader expects to have that information when reading the methods. Line 145. Please change the term Immunogenicity. As mentioned above, it does not adequately define what is described in that section (methods to assess the immune response). Use the term “immunity assessment” or “Immune response assessment” .
Immunnoassay tests (lines 173-180), are insufficiently described. What was the test that was used? Who was the manufacturer? What were the detection limits and the cut offs? If it is an "in home made" assay, the reagents used and the protocol applied must be described exactly. Sample size: only the size of the global sample has been calculated but it does not say what the minimum size of each of the cohorts should be, if it is assumed that they should be 1/3 of this number, obviously a serious problem arises in cohort 2 between 74 (theoretical) and 12 (real cohort size).
Introduction: in the first sentence it should be said that SARS-CoV2 is responsible for SARS, it is an incomplete definition, it should be said that it is responsible for COVID-19 or it is responsible for a form of SARS and COVID19.
Author Response
A point-by-point response to the reviewer’s comments is provided as an attachment. The added/modified parts have been highlighted in yellow in the revised mansucript.

Reviewer 3 Report
The work presented by Demirhindi et al. is about the application of a "mixed" vaccination strategy for the prevention of Covid-19 disease. Although the topic is interesting, in my opinion it comes a bit late, since vaccination against Sars-Cov-2 started 1 year ago and, nowadays, strategies about mixed vaccination are already applied. Beside that, methods used does not support conclusion; the main problem is the numerosity of the three cohort groups, in particular the fact that cohort II is composed of only 17 people (which decrease to 12, if individuals with a history of COVID-19 are removed, as stated by Table 2), while cohort III is ten times bigger. Morevoer, at line 184, what is the purpose of check normality of the data and the using a non-parametric test, which is less powerful than a parametric one as an ANOVA? Given the distribution of data as seen in figure 1, with a lot of outliers, 17 (or 12, is not clear the final numerosity of cohort II group) is not an enough number for such kind of statistical analyses.
Author Response
A point-by-point response to the reviewer’s comments is provided as an attachment. The added/modified parts have been highlighted in yellow in the revised mansucript.
Question 1 The work presented by Demirhindi et al. is about the application of a "mixed" vaccination strategy for the prevention of Covid-19 disease. Although the topic is interesting, in my opinion it comes a bit late, since vaccination against Sars-Cov-2 started 1 year ago and, nowadays, strategies about mixed vaccination are already applied. Beside that, methods used does not support conclusion; the main problem is the numerosity of the three cohort groups, in particular the fact that cohort II is composed of only 17 people (which decrease to 12, if individuals with a history of COVID-19 are removed, as stated by Table 2), while cohort III is ten times bigger. Morevoer, at line 184, what is the purpose of check normality of the data and the using a non-parametric test, which is less powerful than a parametric one as an ANOVA? Given the distribution of data as seen in figure 1, with a lot of outliers, 17 (or 12, is not clear the final numerosity of cohort II group) is not an enough number for such kind of statistical analyses.
Reply 1 First, we would like to thank the reviewer for the time and effort.
As observed, the starting of immunisation did not result in ending the pandemic. Otherwise the number of cases in the whole world would not have increased so much. We aimed to contribute to scientific evidence about assessing the right vaccination strategy. This is particularly important in countries lacking adequate vaccine supply, where inactivated vaccines are prevalent or the only available option. The mixed vaccine strategy is expected to provide the highest benefit in such countries.
Reply to questions realted to the methodology
When the study started, all healthcare workers received the first dose with the inactivated vaccine as the inactivated vaccine was the only option in the country. After the mRNA vaccine was provided, people preferred booster doses according to their own preferences or refused to have a booster dose. The low number of participants is a limitation noticed also by us. However, our number was determined by the power analysis applied before the beginning of the study and people were included randomly. This study is not an experimental or trial study, it is an observational study, so the participation in the experimental groups and the withdrawal of the participants from the study are not controlled by the researchers. It is a disadvantage of observational studies. Despite all this, in cases where the assumptions were not met in the statistical analysis, non-parametric tests were preferred and necessary adjustments were made. As a general rule, according to the central limit theorem, the number of n=30 sufficient for normal distribution was provided for each vaccine cohort (group 1=50, group 2=117, and group 3=168 individuals) statistically at least 10 individuals in each group analyzes were satisfactory. The main findings of the study appear to have met these assumptions. We expect the evaluation of the manuscript in this context.
Round 2
Reviewer 2 Report
Line 269 is incomprehensible, please clarify what you mean Among the limitations of the study should be included the disproportion between the number of components of the three cohorts, especially the fact that in the anti S-RBD IgG study, cohort 2 only had 12 people versus 137 in cohort 3. In the discussion, it should be clarified that the fact that there is a higher proportion of anti-S-RBD response in those reimmunized with BNT does not imply that these patients are more protected, since this study does not assess protection, but levels of antibodies against that antigen.
Author Response
Reviewer #2 PLEASE NOTE: Changes in the manuscript text have been highlighted in YELLOW. Question 1: Line 269 is incomprehensible, please clarify what you mean.
Reply 1: Line 269 Sorry for the mistake. The reply was adequate, but the translation was forgotten. It has been corrected as “…p values in post-hoc analyses: e p<0.001, f p<0.001, g p=0.264, h p=0.002, j p<0.001, k p<0.001…”
Question 2: Among the limitations of the study should be included the disproportion between the number of components of the three cohorts, especially the fact that in the anti S-RBD IgG study, cohort 2 only had 12 people versus 137 in cohort 3.
Reply 2: As suggested by the reviewer the following phrase has been added between lines 364-368: The limitations of the study can be noted as cellular immunity response was not assessed and sampling size was not large. As the study was observational and the selection of the third dose was left to the free will of the individuals (without any intervention by researchers), the proportional parity in the number of participants in three cohorts could not be provided.
Question 3: In the discussion, it should be clarified that the fact that there is a higher proportion of anti-S-RBD response in those reimmunized with BNT does not imply that these patients are more protected, since this study does not assess protection, but levels of antibodies against that antigen.
Reply 3: As suggested by the reviewer the following phrase has been added between lines 353-356: It should be noted that the fact that there is a higher proportion of anti-S-RBD response in those reimmunised with BNT does not imply that these individuals are more protected, since this study does not assess protection, but levels of antibodies against the antigen.
Reviewer 3 Report
The author provides an answer to both the point I pointed. I found partially satisfying the answer number 1, since It is unrealistic to suppose that countries without an "adequate vaccine supply" has a variety of available vaccines so big to choose the best vaccination strategy. However, I recognize the merit to demonstrate that one vaccination strategy is better than another, so contries limited on vaccine supply and so without possibility to choose can be more aware of the risks/benefit of the vaccination strategy employed.
The main problem lies on my second point about the numerosity of the three cohort groups. I understand that your study is only observational and that you do not have any control on the number of participants. This is a clear limit and it is so big to hamper the whole study and prevent its publication. The number you provided in the answer are totally in disagreement with those provived in the articole: the numerosity of the cohort written in the answer is group 1=50, group 2=117, and group 3=168 individuals, but in the paper both in table 1 and lines 198-204, numerosity is group 1=50, group 2=17, and group 3=168 individuals. So the problem of having a group 2 of only 17 individuals (which later further decrease to 12, having removed those individuals which reported a COVID-19 infection history before the first dose) in comparison with a gruop more than 10 times bigger still remains and, for this, the work should not be published.
Author Response
Reviewer #3 PLEASE NOTE: Changes in the manuscript text have been highlighted in YELLOW.
Question: The author provides an answer to both the point I pointed. I found partially satisfying the answer number 1, since It is unrealistic to suppose that countries without an "adequate vaccine supply" has a variety of available vaccines so big to choose the best vaccination strategy. However, I recognize the merit to demonstrate that one vaccination strategy is better than another, so contries limited on vaccine supply and so without possibility to choose can be more aware of the risks/benefit of the vaccination strategy employed.
The main problem lies on my second point about the numerosity of the three cohort groups. I understand that your study is only observational and that you do not have any control on the number of participants. This is a clear limit and it is so big to hamper the whole study and prevent its publication. The number you provided in the answer are totally in disagreement with those provived in the articole: the numerosity of the cohort written in the answer is group 1=50, group 2=117, and group 3=168 individuals, but in the paper both in table 1 and lines 198-204, numerosity is group 1=50, group 2=17, and group 3=168 individuals. So the problem of having a group 2 of only 17 individuals (which later further decrease to 12, having removed those individuals which reported a COVID-19 infection history before the first dose) in comparison with a gruop more than 10 times bigger still remains and, for this, the work should not be published.
Reply: We agree that the difference in the numerosity of the cohorts is a limitation. Therefore, we added this limitation between lines 364-368: The limitations of the study can be noted as cellular immunity response was not assessed and the sampling size was not large. As the study was observational and the selection of the third dose was left to the decision of the individuals (without any intervention by researchers), the proportional parity in the number of participants in three cohorts could not be provided.
But, we do not agree that this prevents the publication of the article. The ref #12 (Zhang, Y.; Zeng, G.; Pan, H.; Li, C.; Hu, Y.; Chu, K.; Han, W.; Chen, Z.; Tang, R.; Yin, W.; et al. Safety, tolerability, and immunogenicity of an inactivated SARS-CoV-2 vaccine in healthy adults aged 18–59 years: a randomised, double-blind, placebo-controlled, phase 1/2 clinical trial. Lancet Infect. Dis. 2021, 21, 181–192, doi:10.1016/S1473-3099(20)30843-4) is the phase 1/2 clinical trial for the inactivated vaccine which we utilised for sample size calculation. As can be seen, the number in cohorts are as low as 12 or 24. This has been published considering the numbers satisfactory by a recognised journal and tens of countries initiated the vaccination with this inactivated vaccine taking these numbers as reference. On the other hand, appropriate statistical methods have been used to ascertain the significance of the results found with appropriate corrections. Non-parametric tests are for these cases. Statistical significances have been assessed. Therefore, we think the manuscript merits publication.
Besides this, the proportions of the cohorts are representative of the community. It reflects the distribution of the vaccine preference at the hospital where the study was conducted, and the distribution in the country as the preferences are almost similar among health workers suggested to the same conditions, too. To summarise again shortly; only the inactivated vaccine was available in Turkey when the study was started. Before all public groups, the health workers were given the chance to be vaccinated, and of course with the inactivated vaccine. Then, when BNT became available, almost all health workers shifted to BNT. It is very rare to find a 3-dose-inactivated vaccine (CV) preference in the country, especially among health workers except those having allergic/chronic disease comorbidities. The fact that the reviewer finds this hard to imagine. There are no studies from developed countries
regarding the inactivated vaccine, which is very extensively used in the world. This is one reason why this article should be published. The findings need to be shared with the scientific community and policymakers for the benefit of the related communities, but also the whole world as the pandemic is not limited to a certain area of the world. If we cannot stop the spread in the developing world, it will not be eliminated in the developed world. And if we want to eliminate the disease (eradication appears not to be possible for now), we should take action quickly.
Please consider that the main outcome of the study is showing the superiority of HETEROLOGOUS vaccination. It is a quasi-new concept and it should be discussed and evaluated for combat against COVID-19. It is true that the low number of participants in cohort 2 is a limitation and shades the power of the study. However, this does not shade the finding that the heterologous strategy is more effective compared to the homologous one. The statistical significance of the finding is guaranteed by using appropriate (non-parametric) techniques. It has been emphasised that the third dose of inactivated vaccine is also beneficial in providing a greater immune response. On the other hand, extreme outliers are not observed in cohort 2 when compared to other cohorts.
These limitations have been inserted in the text between lines 364-368 as also mentioned above.
As a conclusion; these limitations do not render the study unpublishable which is expected to significantly add to the scientific knowledge and the literature by emphasising the benefit of heterologous vaccination in the context of immune response by the inactivated vaccine and mRNA vaccine. It is also intended to close the gap observed in the article’s numerosity regarding the inactivated vaccine. To generalise a single finding to the whole of the manuscript will shade the contribution of the study to the scientific community.
Sorry for the mistake. The reviewer is right. It has been a keyboard error. The number of participants in group 2 has been corrected at line 157 as: Cohort-II (CV/CV/CV) included those who received three doses of CoronaVac (n=17)